# Enrichment of Ewe’s Milk with Dietary n-3 Fatty Acids from Palm, Linseed and Algae Oils in Isoenergetic Rations

**DOI:** 10.3390/ani12131716

**Published:** 2022-07-02

**Authors:** Teresa Manso, Beatriz Gallardo, Paz Lavín, Ángel Ruiz Mantecón, Carmen Cejudo, Pilar Gómez-Cortés, Miguel Ángel de la Fuente

**Affiliations:** 1Departamento de Ciencias Agroforestales, Universidad de Valladolid, 34004 Palencia, Spain; tmanso@agro.uva.es (T.M.); beatriz.gallardo.garcia@uva.es (B.G.); 2Instituto de Ganadería de Montaña (CSIC-ULE), Grulleros, 24346 León, Spain; paz.lavin@eae.csic.es (P.L.); mantecon@eae.csic.es (Á.R.M.); 3Instituto de Investigación en Ciencias de la Alimentación, CSIC-UAM, Nicolás Cabrera 9, 28049 Madrid, Spain; carmencejudog@gmail.com (C.C.); ma.delafuente@csic.es (M.Á.d.l.F.)

**Keywords:** dairy fat, omega-3 fatty acids, sheep milk, linseed oil, calcium salts, algae oil

## Abstract

**Simple Summary:**

The aim of this research was to evaluate the effects of three lipid supplements in ovine diets on milk healthy fatty acid composition. Rations provided a source of α-linolenic acid, either protected or unprotected from ruminal biohydrogenation (linseed oil and Ca-salts of linseed oil) or long-chain n-3 fatty acids (algae oil). Algae oil supplementation generated the highest levels of n-3 fatty acids in ewe’s milk. However, dry matter intake, milk yield and fat content declined with the inclusion of algae oil in the diet. The fatty acid profiles of milk from both linseed oil supplemented rations did not significantly differ, and both were effective to diminish saturated fatty acids.

**Abstract:**

Increasing the levels of n-3 fatty acids (FA) in dairy products is an important goal in terms of enhancing the nutritional value of these foods for the consumer. The purpose of this research was to evaluate the effects of linseed and algae oil supplements in ovine isoenergetic diets on healthy milk fatty acid composition, mainly n-3. Seventy-two Churra dairy ewes were divided and randomly assigned to four experimental treatments for 6 weeks. The treatments consisted of a TMR (40:60 forage:concentrate ratio) that varied according to the inclusion of different types of fat (23 g/100 g TMR): hydrogenated palm oil (control), linseed oil (LO), calcium soap of linseed oil (CaS-LO) and marine algae oil (AO). The most effective lipid supplement to increase n-3 FA in milk was AO. 22:6 n-3 and total n-3 PUFA content increased from 0.02 and 0.60% (control) to 2.63 and 3.53% (AO), respectively. All diets supplemented with n-3 FA diminished the content of saturated FA in milk and its atherogenic index, while the levels of *trans*-11 18:1 and *cis*-9 *trans*-11 18:2 significantly increased. Overall, the enhancement of n-3 FA in ewe’s milk would be advantageous for the manufacture of nutritionally improved cheeses.

## 1. Introduction

The nutritional importance of n-3 fatty acids (FA), such as 18:3 (α-linolenic acid, ALA), 20:5 (eicosapentaenoic acid, EPA), 22:6 (docosahexaenoic acid, DHA) and more recently, 22:5 (docosapentaenoic acid, DPA), in human nutrition is widely documented [1]. They are essential for normal physiological function and are associated with multiple health benefits [2]. This evidence has created great interest in the development of functional foods that incorporate n-3 FA in their composition [3]. The amounts of n-3 FA in dairy fat from ruminants are very low [4,5,6]. This fact, together with the relevant contents of saturated (SFA) and *trans* FA, has caused some nutritionists to advise against the consumption of milk fat and whole dairy products.

Most of the sheep’s milk in the world is processed into cheese. It is well known that cheese making and ripening do not modify the FA profile of dairy fat; thus, the most suitable strategy to enhance the FA composition in ewe cheese is to increase healthy FA in raw milk [7,8]. In this line, there is currently a great interest in adding value to sheep milk by reducing its SFA content while increasing the levels of n-3 FA.

The content of n-3 FA in milk fat can be naturally enhanced by supplementing the diet of ruminants with a lipid source enriched in polyunsaturated fatty acids (PUFA) [9]. The inclusion of linseed in rations [8,10] or the use of diets based on fresh pastures [11] have been shown to increase the ALA content in ewe’s milk. However, ALA increases were modest and its content did not exceed 2% of total milk FA. Regarding long-chain n-3 FA, EPA and DHA increases have been reported when adding algae oils of marine origin to lactating ewe rations [12,13,14]. Microalgae stand out as a sustainable source of DHA and their inclusion in ruminants’ diet enhances n-3 FA content in animal products [15]. Notwithstanding, the percentage of n-3 PUFA reached in milk fat seldom exceeded 1% [12,14,16].

The limited incorporation of n-3 FA in milk fat is mainly related to the high biohydrogenation (BH) rates of n-3 FA in the rumen, which drastically reduce their intestinal absorption, limiting their availability to the mammary gland and milk fat secretion [17]. Minimizing ruminal BH of PUFA is the major challenge in dietary supplement formulations, which try to enhance the post ruminal supply of PUFA. A proposed fat protection technology is the use of calcium salts (Ca-salts) of selected FA that would reduce the negative effects of fat on ruminal fermentation [18]. As linseed oil is rich in ALA, Ca-salts of linseed oil would be a suitable option to increase n-3 FA concentration in milk. To the best of our knowledge, no information is available comparing the effects of free linseed oil with Ca-salts from linseed oil as feed ingredients for dairy sheep. On the other hand, few data are available on the BH pathways of DHA and the in vivo effects of the inclusion of unprotected microalgae oils in the diet of ewes [19]. Since the BH of dietary PUFA is usually incomplete, we hypothesized that the incorporation of high levels of microalgae oil may enhance the outflow of FA to reach the duodenum and, after absorption, n-3 FA could be available in the mammary gland and incorporated into milk fat. Some authors have pointed out that lipids can reduce the activity or proliferation of rumen bacteria, modulating the pathways of fatty acid BH in the rumen and allowing PUFA to pass the rumen and be absorbed at the intestinal level [20,21]. Likewise, it has been pointed out that algae can produce toxins and reduce the activities of some microorganisms, inhibiting BH processes at the ruminal level and increasing the possibilities of reaching the intestine for absorption.

It is difficult to select the appropriate lipid supplement to increase n-3 FA content in milk and improve the nutritional characteristics of dairy fat, mainly because the experimental conditions of the available literature vary greatly (e.g., using unequal basal diets, the type of n-3 supplements, amounts added to the rations, lactation period, animal parity…). Therefore, the objective of this work was to evaluate the in vivo effects of isoenergetic diets providing the same amount of fat but differing in the n-3 FA source (either rich in ALA (linseed oil or Ca-salts of linseed oil) or long-chain n-3 FA (algae oil) on animal performance and healthy milk FA composition.

## 2. Materials and Methods

### 2.1. Animals and Dietary Treatments

Seventy-two Churra dairy ewes (58.0 kg ± 8.11 kg of body weight) in the sixth week of lactation (40.2 ± 2.74 days of lactation) were divided into twelve homogenous lots of 6 animals each, and randomly assigned to 4 experimental treatments (3 lots per treatment). The experiment was carried out in a commercial sheep farm. All three lots were fed at the same time on the farm from May to June. The trial lasted 5 weeks and all animal handling practices followed the recommendations of the European Council Directive 2010/63/EU for the protection of animals used for experimental and other scientific purposes. The experimental procedures were approved by the Institutional Animal Care and Use Committee of the University of Valladolid.

Treatments consisted of a total mixed ration (TMR, 40:60 forage:concentrate ratio) that varied according to the inclusion of different types of fat (2.3 g/100 g TMR): hydrogenated palm oil (control), linseed oil (LO), calcium soap of linseed oil (CaS-LO) and marine algae oil (AO). Experimental diets were formulated to be isoenergetic and isoproteic and were supplied ad libitum twice a day (9:00 and 18:00). Fresh water was always available. Ingredients and chemical composition of the experimental diets are presented in Table 1. During a two-week adaptation period (before the beginning of the trial), all animals received the same TMR composed of dehydrated alfalfa hay (37%), barley straw (9%), soybean meal (14%), whole corn grain (11%), oat grain (9%), whole barley grain (8%), beet pulp (7%), molasses (4%) and vitamin-mineral premix (1%); 93.2% DM; ash, 6.9% DM; neutral detergent fiber, 34.9% DM; acid detergent fiber, 22.8% DM; crude protein, 17.5% DM; and ether extract, 2.2% DM.

During the whole experimental period, dry matter intake (DMI) was recorded weekly. The amounts of offered diet and refusals were weighed daily and samples of each lot of ewes were pooled weekly for subsequent analyses. The chemical composition of feed and refusal samples was determined by using the procedures described by AOAC International [22]. Dried feed samples were analyzed for NDF and ADF using filter bag equipment (Ankom Technology Corp., Fairport, NY, USA).

### 2.2. Milk Sampling and Composition

Ewes were machine-milked twice a day (8:00 and 17:00 h) during the whole experimental period using a 2 × 18 low-line Casse system milking parlor (Alfa-Laval Iberia, S.A., Madrid, Spain) with 12 milking units and 2 milkers. The milking machine was set to provide 180 pulsations per minute in a 50:50 ratio with a vacuum level of 36 kPa. After the two-week adaptation period, individual ewe milk production was recorded weekly and samples were taken in milk collection jars. One subsample of milk was kept at 4 °C until it was analyzed for fat and protein content in accordance with International Dairy Federation recommendations [23] using a MilkoScan-400 analyzer (Foss Electric, Hillerød, Denmark). Milk samples were collected from individual ewes every two weeks of the experimental period and stored at −80 °C for subsequent FA analysis.

### 2.3. Fatty Acid Analysis

Lipids extraction was carried out by two consecutive centrifugations from 30 mL of raw milk using the method proposed by Luna et al. [24]. Dried dairy fat was stored in amber vials, blanketed with a stream of N_2_ and kept at –20 °C until analysis. Fatty acid methyl esters (FAME) were prepared by base-catalyzed methanolysis of glycerides with KOH in methanol [25]. Each transesterification mixture (0.2 mL) was reacted with 25 mg of milk fat at room temperature.

Two Agilent models (6890 N Network Gas Chromatograph and 7820A GC System) (Agilent, Palo Alto, CA, USA) equipped with autoinjectors, fitted with flame-ionization detectors and CP-Sil 88 fused silica capillary columns (100 m × 0.25 mm i.d., Varian, Middelburg, The Netherlands), were used to determine the FAME profile. The injector and detector temperature was 250 °C. Helium was the carrier gas at an inlet pressure of 193.9 kPa and a split ratio of 1:100. GC analysis was performed with two complementary oven temperature programs. In the first GC program, initial oven temperature was set at 45 °C. After 4 min, it was raised at 13 °C/min to 165 °C and held for 35 min, then increased to 215 °C at 4 °C/min and maintained for 30 min. In the second GC program, the initial oven temperature was set at 195 °C and, after 45 min, it was raised at 4 °C/min to 215 °C and held for 30 min. The combination of both GC analyses allowed the resolution of all *iso* and *anteiso* FA present in dairy fat, which was not possible with a single GC run.

The identification of FAME was carried out by comparison with commercial standard mixtures from Nu-Chek Prep Inc. (Elysian, MN, USA) and analogous milk fat samples from previous research. Individual FAME quantification was performed using response factors calculated from a milk fat with a certified composition (CRM 164; European Community Bureau of Reference, Brussels, Belgium) according to ISO-IDF [26].

### 2.4. Statistical Analysis

Data regarding DMI, milk production and chemical and FAME composition were analyzed using the MIXED procedure of SAS (version 9.4; SAS Institute Inc., Cary, NC, USA). The model took into account the fixed effects of dietary treatment (D), week of sampling (T), and their interaction (D × T). Time on diet was considered a repeated factor, lot as a blocking factor and animal nested within treatment was subjected to a compound symmetry-covariance structure. Significant differences were declared at *p* < 0.05. As the interaction was not significant in most cases, only means for the principal effects are presented in the tables.

## 3. Results and Discussion

### 3.1. Animal Performance and Milk Composition

Table 2 shows the data of DMI, milk yield and milk composition of ewes fed the four assayed diets. Average DMI changed significantly (*p* < 0.01) when the rations incorporated n-3 lipid supplements and all milk-related parameters also with time on diet. In contrast, the diet × time interaction was less pronounced.

AO supplementation strongly reduced DMI (*p* < 0.01) and milk yield (*p* < 0.01), when compared to the control, LO and CaS-LO diets. Previous studies have reported decreases in feed intake when marine algae were administered to cows [27], goats [28] and dairy sheep [13]. The drop in DMI caused by AO treatment may be justified by some aspects related to long-chain PUFA, such as the lower acceptability of AO due to its palatability [12] or reductions in microbial activity and fiber fermentation. In fact, disturbance of rumen fermentation through very long chain PUFA could explain most of the reduced DMI. On the other hand, marine lipids could also be toxic to ruminal microbiota and, therefore, negativezly affect ruminal fermentation processes and animal feed consumption [29]. The high dose of unprotected AO administered in the current experiment (2.3%) should be taken into account, since smaller doses (<1.5%) of marine algae did not decrease DMI in dairy sheep [16]. The lower DMI observed in ewes from AO (*p* < 0.001) would explain the reduction of milk yield in animals following this treatment (*p* < 0.001).

In comparison to the control diet, average DMI and milk yield did not change (*p* > 0.05) in ewes fed diets supplemented with LO and CaS-LO, and no remarkable changes were observed between LO and CaS-LO rations. Previous experiments [7,30] have reported that ewes readily accept LO and that feeding up to 3% of LO (DM basis) would have no effect on DMI and milk yield. Similarly, Côrtes et al. [31] observed no effect on DMI or milk production when cows were fed with whole flaxseed or calcium salts of flaxseed oil. In the present experiment, although milk yield did not significantly differ among LO and CaS-LO, numerically, CaS-LO treatment showed the highest milk yield. It has been pointed out that improvements in milk yield could be accounted for by the greater digestibility of fiber and digestible energy content of the diet when calcium soaps of fatty acids were fed in comparison to free oil [32]. The controversies between studies on animal performance affected by oil and calcium soaps could also be explained by differences in the lipid dosage [33].

The lowest milk fat yield and fat percentage were found in the AO treatment (Table 2). However, the reduction of milk fat percentage was limited despite the high level of AO (2.3%) included. Decreased fat contents in milk caused by algae supplementation has been previously reported in several studies [14,16,27]. The reduction in milk fat content has been associated with a negative energy balance as a result of low feed intake or to a low fat syndrome related to the action of EPA and DHA on rumen microbiota, favoring alternative BH pathways that produce antilipogenic metabolites [34]. In the present research, BH intermediates that could explain marine lipid-induced milk fat depression, such as *trans*-10 18:1, remained low. Therefore, their roles as milk fat synthesis inhibitors could not be justified and the reduction in dairy fat production would suggest that other FA or mechanisms might have an additional role in milk fat secretion [35]. In this line, the low fat syndrome of AO animals could be caused by a reduced synthesis of *cis*-9 18:1. Oleic acid has been proposed as one of the principal FA responsible for the maintenance of milk fat globule fluidity in the mammary gland and, consequently, for their secretion [16,36]. On the other hand, several BH intermediates derived from AO could also reduce the expression of enzymes involved in de novo FA synthesis in the mammary gland, as it has been reported in molecular studies [34]. The role of ruminal volatile FA, such as in the decrease in ruminal acetate, also deserves attention in this matter, as de novo FA synthesis is stimulated by acetate and marine oils may reduce its production [35]. The present results show that long-chain PUFA may be able to inhibit acetate producing bacteria and therefore the de novo lipogenesis as observed in many in vivo and in vitro investigations [31]. Apart from that, the concentration effect linked to the decrease in milk yield could explain the limited milk fat reduction despite the high level of AO, prevailing over the effect of AO on de novo FA synthesis [13].

The addition of fat to ruminant diets generally reduces the protein content of milk due to an increase in milk production [37]. However, in the present study, milk yield was reduced in AO diets and a concentration effect could explain the observed increase in milk protein and also the reduction in lactose concentration. These findings are in line with previous research in lactating cows [27], goats [28] and sheep [12,13] fed with *Schizochytrium* sp. diets.

No remarkable changes were observed in daily fat, protein, lactose or total solids production among control and LO and CaS-LO treatments (Table 2), due to the lack of changes in DMI and milk yield. LO and CaS-LO diets increased (*p* < 0.01) milk fat concentration when compared to the addition of hydrogenated palm oil (control). The intestinal absorption of fat consumed by ewes following the LO and CaS-LO treatment could be higher than in the control and this is probably the reason that explains its higher milk fat content. Changes in total solids in milk reflected the main changes in fat, protein and lactose content (Table 2).

### 3.2. Saturated Fatty Acid Profile

Table 3, Table 4 and Table 5 show the detailed FA profile of milk from ewes fed the four diets assayed. Dairy fat composition was substantially modified when rations incorporated different n-3 FA sources. In contrast, the effects associated with time on diet or the interaction diet × time were less prominent.

Total non-branched SFA levels in milk were significantly reduced (*p* < 0.01) due to the replacement of SFA with n-3 FA in diets. Linseed oil treatment showed the lowest contents (65.07%), followed by CaS-LO (68.15%) and AO (69.87%). The steepest declines were observed for even FA from 10 to 18 C atoms. LO and CaS-LO feeding diminished (*p* < 0.01) 12:0, 14:0 and 16:0, whose sum was reduced by 26% (LO) and 20% (CaS-LO), when compared to the control diet (Table 3). This reduction was not as drastic with AO (9% drop) but this supplement showed a critical effect on 18:0 milk content.

The group of medium-chain SFA that includes 10:0, 12:0, 14:0 and 16:0 is mostly synthesized de novo in the mammary gland. It is well established that increases in the supply of long-chain PUFA alter the synthesis of those medium-chain SFA [37,39]. Dietary PUFA compete for esterification with synthesized medium-chain SFA in the mammary gland and the accumulation of such FA may lead to feedback inhibition of lipogenic enzymes [40].

The low levels of 18:0 in AO diet milk is related to the inhibitory effect that oils from marine origin could exert on the last step of ruminal BH: the formation of 18:0 from MUFA in the rumen. Altogether, the decrease in 18:0 and the increase in *trans*-18:1 in milk from ewes fed the AO diet reflect the action of the long chain n-3 PUFA present in marine algae, mainly DHA [41], on *trans*-18:1 ruminal accumulation [42].

The hypercholesterolemic SFA (12:0 + 14:0 + 16:0) was significantly diminished (*p* < 0.01) in the three n-3 FA supplemented diets, but in higher proportion in the LO diet (Table 3). A major concern of consuming whole milk is the high proportion of SFA in dairy fat and, thus, the observed reduction of hypercholesterolemic SFA should be considered positive from a nutritional standpoint. In this line, it has been reported that reducing SFA at the expense of increasing monounsaturated fatty acids (MUFA) in dairy products would be an effective strategy to improve endothelial function and LDL cholesterol levels in patients with cardiovascular risk [43].

The proportion of branched-chain fatty acids (BCFA) was about 1.5% (Table 3). BCFA are an emerging group of bioactive FAs sparking growing research interest within the scientific community due to their biological effects and potential pro-health benefits [44]. Because BCFA are principally derived from rumen bacteria, milk and dairy products pose unique dietary sources. Overall, in the current research, only CaS-LO decreased the total content of this FA group in milk fat. Regarding the most abundant individual BCFA, *iso* 16:0 and *anteiso* 17:0 contents were not affected by lipid supplementation (Table 3) whereas *anteiso* 15:0 levels were higher in AO experimental treatment.

### 3.3. Monounsaturated Fatty Acid Profile

Table 4 shows the contents of monounsaturated FA (MUFA) in milk fat obtained with the assayed diets. As can be seen, total MUFA with the *cis* configuration differed among treatments. Percentages in the LO and CaS-LO treatments were higher (*p* < 0.01) than the control, whereas a substantial reduction was observed when ewes were fed with AO. These trends should be principally attributed to the behavior of *cis*-9 18:1 (oleic acid), quantitatively the most abundant 18:1 isomer in dairy fat. An increase of oleic acid in milk tends to be closely linked to the levels of 18:0. The 18:0 generated in the rumen, after complete BH of unsaturated lipids, can be subsequently converted into *cis*-9 18:1 in the mammary gland via ∆9-desaturase, contributing to increase in the contents of oleic acid in milk.

In contrast, oleic acid levels were reduced by half when AO was included in ewe rations (Table 4). As mentioned above, this decrease would be indirectly derived from the inhibitory action of long-chain n-3 PUFA present in AO, mainly DHA, on the ruminal microorganisms involved in the conversion of MUFA to 18:0. Subsequently, the low supply of 18:0 to the mammary gland would reduce the formation of oleic acid in this organ.

Other 18:1 isomers such as *cis*-12 and *cis*-15 experienced a strong rise with LO diets, multiplying their contents in milk fat. Both 18:1 isomers have been documented as intermediates of the ALA BH processes [4,6]. In comparison, the presence of AO, a lipid substrate very poor in ALA (Table 1), did not increase the content of these 18:1 *cis* isomers in ewe milk fat.

Regarding *trans* FA, 18 C atoms isomers were the most prominent in all diets. The rest of the *trans* MUFA were measured in much smaller quantities (Table 4). Overall, total *trans* MUFA detected in PUFA supplemented diets quadrupled the control milk values, without significant differences among them. However, the behavior of the most abundant isomers, *trans*-10 and *trans*-11 (vaccenic acid, VA), was influenced by the type of lipid supplement. LO and CaS-LO multiplied the VA content by eight, whereas AO sharply increased the level of this *trans* isomer from 0.42 (control) to 5.20%. Concerning *trans*-10 18:1, only the LO diet significantly modified (*p* < 0.01) the content of this isomer with respect to the control value.

The increment of VA in ewe milk fat is quite characteristic of diets enriched in ALA [7,8,11] and, therefore, it was expected. This *trans* MUFA is an important intermediate in the process of ALA BH [4,6]. Regarding the VA levels detected in AO diet milk, its presence may be associated to the action of the long chain n-3 PUFA present in AO, mainly DHA, a potent inhibitor of *trans*-18:1 ruminal final reduction, which induced the accumulation of VA in the rumen [45].

Apparently, the gain of a *trans* FA should be considered negative from a nutritional point of view. Nevertheless, since more than a decade ago, a number of positive health effects have been ascribed to VA [5]. The greatest interest in increasing VA concentration in dairy fat has come from its role as substrate for the endogenous synthesis of *cis*-9 *trans*-11 18:2 (rumenic acid, RA) via ∆9-desaturase, not only in the ruminant mammary gland but also in human tissues [45]. RA is one of the most relevant bioactive compounds present in milk fat. It exhibits, both in vivo and in vitro, antitumor, anti-atherosclerosis and antidiabetic effects, and also modulates the immune system [5]. As can be seen in the current research, those diets inducing the largest accumulation of VA, AO principally, also resulted in the highest RA concentrations.

Other *trans* 18C MUFA did not experience changes with the different lipid supplements or their increases were moderated in comparison to VA. The lack of an increase in *trans*-10 18:1 when supplementing AO in the current study (Table 4) is particularly remarkable because this FA has been associated with the risk of cardiovascular disease in humans [46]. The main reason for the absence of *trans*-10 18:1 changes may be linked to the source of lipid supplementation. *Trans*-10 18:1 milk content has been reported to increase in ewes when supplementing ewe diets with a source of linoleic acid [14,16] and its concentration was low (LO and CaS-LO) or very low (AO) in our assayed lipid supplements (Table 1). *Trans*-10 18:1 was only augmented in the LO treatment by 0.25 to 0.77%, very far from the percentage reached by VA.

It is also striking that no differences were observed in the contents of any *trans* 18C MUFA between LO and CaS-LO diets. These results would be evidence supporting the poor efficiency of calcium salts to protect ALA present in LO when they pass through the digestive tract.

### 3.4. n-3 Fatty Acid Profile

Table 5 reports the percentages of the PUFA obtained in milk fat with the four diets assayed. The highest levels of ALA were found in LO and CaS-LO supplemented diet milks, 0.82 and 0.70%, respectively, doubling the values of the palm-oil-supplemented diet animal milk samples. In comparison, ALA contents in control and AO treatments did not significantly differ. These results are a clear consequence of the presence of ALA in the rations (Table 1). Adding either linseed or linseed oil, not marine substrate which is poor in this FA, has been demonstrated as the best way to increase ALA in ovine dairy products [7,8].

**Table 5 animals-12-01716-t005:** Polyunsaturated fatty acid (PUFA) profile (g/100 g of total fatty acid methyl esters) of milk fat from ewes fed with 2.3% of different lipid supplements. ALA (α-linolenic acid); DHA (22:6 n-3); DPA (20:5 n-3); EPA (22:3 n-3).

	Diets ^1^					Probability ^3^		
Item	Control	LO	CaS LO	AO	SED ^2^	D	T	D × T
*trans*-11 *trans*-15 18:2	0.02 ^b^	0.02 ^ab^	0.04 ^a^	0.03 ^b^	0.003	0.004	0.020	0.347
*trans*-9 *trans*-12 18:2	0.01 ^d^	0.12 ^a^	0.08 ^b^	0.03 ^c^	0.009	<0.001	0.132	0.186
Other *trans trans* 18:2	0.04 ^b^	0.12 ^a^	0.12 ^a^	0.04 ^b^	0.006	<0.001	0.184	0.100
*cis*-9 *trans*-13 18:2	0.21 ^b^	0.62 ^a^	0.62 ^a^	0.16 ^b^	0.031	<0.001	0.039	0.103
*trans*-8 *cis*-13 18:2	0.12 ^c^	0.25 ^a^	0.21 ^b^	0.05 ^d^	0.017	<0.001	0.016	0.100
*cis*-9 *trans*-12 18:2	0.04 ^b^	0.08 ^a^	0.08 ^a^	0.04 ^b^	0.008	<0.001	0.663	0.730
*trans*-9 *cis*-12 18:2	0.01 ^b^	0.03 ^a^	0.03 ^a^	0.03 ^a^	0.002	<0.001	0.571	0.771
*trans*-11 *cis*-15 18:2	0.03 ^c^	0.76 ^a^	0.46 ^b^	0.40 ^b^	0.068	<0.001	0.768	0.806
*cis*-9 *cis*-12 18:2	2.05 ^a^	1.65 ^b^	1.90 ^a^	1.25 ^c^	0.099	<0.001	0.826	0.451
*cis*-9 *cis*-15 18:2	0.03 ^b^	0.06 ^a^	0.04 ^b^	0.02 ^c^	0.006	<0.001	0.976	0.245
*cis*-12 *cis*-15 18:2	0.01 ^c^	0.17 ^a^	0.08 ^b^	0.01 ^c^	0.018	<0.001	0.746	0.407
*cis*-9 *trans*-11 18:2	0.32 ^c^	1.78 ^b^	1.61 ^b^	2.68 ^a^	0.211	<0.001	0.315	0.721
TOTAL 18:2	2.88 ^c^	5.65 ^a^	5.26 ^ab^	4.71 ^b^	0.283	<0.001	0.201	0.571
18:3 n-6	0.09 ^a^	0.03 ^b^	0.04 ^b^	0.03 ^b^	0.006	<0.001	0.287	0.263
18:3 n-3	0.40 ^c^	0.82 ^a^	0.70 ^b^	0.30 ^c^	0.051	<0.001	0.103	0.216
*cis*-9 *trans*-11 *trans*-15 18:3	0.04 ^c^	0.06 ^ab^	0.07 ^a^	0.05 ^b^	0.006	<0.001	0.000	0.124
*cis*-9 *trans*-11 *cis*-15 18:3	0.05^b^	0.15 ^a^	0.14 ^a^	0.02 ^b^	0.016	<0.001	0.144	0.205
20:2 n-6	0.03 ^a^	0.01 ^b^	0.02 ^a^	0.02 ^a^	0.002	<0.001	0.359	0.526
20:3 n-6	0.02 ^b^	0.02 ^b^	0.02 ^b^	0.04 ^a^	0.002	<0.001	0.456	0.759
20:3 n-3	˂0.01 ^b^	˂0.01 ^b^	˂0.01 ^b^	0.03 ^a^	0.002	<0.001	0.891	0.975
20:4 n-6	0.14 ^b^	0.08 ^c^	0.09 ^c^	0.34 ^a^	0.017	<0.001	0.994	0.634
22:2 n-6	0.01 ^b^	0.01 ^b^	0.01 ^b^	0.04 ^a^	0.001	<0.001	0.014	0.059
20:5 n-3	0.03 ^b^	0.03 ^b^	0.03 ^b^	0.46 ^a^	0.025	<0.001	0.815	0.956
22:4 n-6	0.02 ^b^	0.01 ^b^	0.01 ^b^	0.10 ^a^	0.007	<0.001	0.498	0.368
22:5 n-6	˂0.01 ^b^	˂0.01 ^b^	˂0.01 ^b^	0.58 ^a^	0.029	<0.001	0.905	1.000
22:5 n-3	0.06 ^b^	0.07 ^b^	0.07 ^b^	0.39 ^a^	0.016	<0.001	0.041	0.037
22:6 n-3	0.02 ^b^	0.02 ^b^	0.02 ^b^	2.63 ^a^	0.149	<0.001	0.869	0.992
TOTAL PUFA	0.90 ^b^	1.32 ^b^	1.23 ^b^	5.03 ^a^	0.228	<0.001	0.799	0.952
TOTAL n-3	0.60 ^c^	1.15 ^b^	1.04 ^b^	3.87 ^a^	0.189	<0.001	0.674	0.924
TOTAL n-6	2.35 ^a^	1.82 ^c^	2.09 ^b^	2.40 ^a^	0.117	<0.001	0.712	0.559
n-6/n-3	3.98 ^a^	1.63 ^c^	2.07 ^b^	0.65 ^d^	0.096	<0.001	0.038	0.411
AI ^4^	4.66 ^a^	2.39 ^d^	2.78 ^c^	3.53 ^b^	0.187	<0.001	0.043	0.477
TI ^5^	3.95 ^a^	2.17 ^c^	2.44 ^b^	1.82 ^d^	0.121	<0.001	0.062	0.714
mg of ALA/100 g of milk	20.4 ^c^	48.2 ^a^	38.0 ^b^	14.6 ^c^	3.23	<0.001	0.089	0.225
mg of DPA/100 g of milk	1.7 ^b^	1.9 ^b^	1.7 ^b^	22.1 ^a^	1.21	<0.001	0.658	0.841
mg of EPA/100 g of milk	3.0 ^b^	4.0 ^b^	3.7 ^b^	19.0 ^a^	0.93	<0.001	0.113	0.928
mg of DHA/100 g of milk	1.1 ^b^	1.3 ^b^	1.3 ^b^	127.5 ^a^	7.25	<0.001	0.792	0.970

^1^ Diets: Control = TMR supplemented with 2.3% of hydrogenated palm oil (PROFAT, Mateos S.L., Valladolid, Spain); LO = TMR supplemented with 2.3% of linseed oil (ECOFLAX FEED OMEGA-3, BTSA, Alcala de Henares, Madrid, Spain); CaS-LO = TMR supplemented with 2.3% of fat from calcium salts of linseed oil (LINOFAT, Nutrion International SLU, Madrid, Spain); AO = TMR supplemented with 2.3% of algae oil (BIOMEGA TECH A 40 FEED, BTSA, Alcala de Henares, Madrid, Spain). ^2^ Standard error of the difference. ^3^ Probability of significant effect of lipid supplement (D), time on diet (T), and their interaction (D × T). ^a–d^ Means with different superscripts differ significantly (*p* < 0.05). ^4^ Atherogenic Index (AI) = (12:0 + 4 × 14:0 + 16:0)/(MUFA + PUFA) [47]. ^5^ Thrombogenic Index (TI) = (C 14:0 + C16:0 + C18:0)/[(0.5 × MUFA) + (0.5 × ∑ n-6) + (3 × ∑ n-3) + (∑ n-3/∑ n-6)] [47].

The similarity in milk ALA contents between CaS-LO and LO diets indicates that Ca-salts of LO were not very useful to protect and increase the levels of this n-3 FA in milk fat. Even more, ALA was higher with unprotected LO (Table 5). The raising of *cis*-9, *trans*-11 *cis*-15 18:3 and *cis*-9 *trans*-11 *trans*-15 18:3 in both ALA enriched diets would be additional evidence supporting this idea. Both 18:3 isomers are intermediates of the BH process of ALA [48]. If the use of Ca-salts were effective to avoid the degradation of ALA in the digestive tract, both 18:3 BH intermediates should have been found in lower percentage in CaS-LO when compared to LO supplemented diet milks.

The presence of AO in the diet of lactating ewes significantly increased (*p* < 0.01) the contents of all long-chain n-3 FA (DPA, EPA and DHA) in milk (Table 5), whereas feedings based on LO did not modify them. The most relevant increase was achieved for DHA (2.63%). Microalgae are the original source of DHA in the marine food chain, and its presence in animal feeds has been considered as a means of enhancing the level of this long-chain n-3 PUFA in foods of animal origin [15,49]. Previous studies have demonstrated that the inclusion of microalgae improved the nutritional properties of ewe´s milk with regard to DHA composition [12,13,16].

Therefore, despite ruminal BH, a proportion of the dietary DHA would have reached the small intestine intact and would have subsequently been absorbed and deposited in the mammary gland when ewes were fed with AO. Several reasons could be behind this non-altered DHA growth (Table 5). Maia et al. [50] suggested that the inclusion of microalgae may exert toxic effects on ruminal bacteria, mainly in the *Butyrivibrio* group, considered to be the dominant species responsible for PUFA BH. Marine algae are a source of phenolic compounds that possess antioxidant and antibacterial properties [51]. It has also been reported that ruminants fed with high tannin levels show greater PUFA contents in their tissues, in comparison to animals fed with plants containing lower levels of these compounds [52].

On the other hand, BH is not the only mechanism that could affect the transfer to milk fat of DHA from the diet. n-3 FA is preferably transported in plasma phospholipids, which limits their transfer to the mammary gland [4]. However, there seems to be some evidence that when the absorbed amount of DHA is high, the capacity of its transport by phospholipids may be saturated and part of the available DHA would be transported as triglycerides, resulting in a higher transfer efficiency from the blood to milk fat [53]. Nevertheless, the maximal proportion of DHA that ewe milk fat can accommodate still remains unclear. Despite the high DHA content of the AO supplement (44% of total FA), increases in milk fat were limited and represented an average transfer efficiency (g in milk/100 g consumed) of only 4.9%, in line with or lower than previous transferences reported in dairy ewes [12,13,14,16].

The second most abundant n-3 FA in the AO diet was EPA, with 0.46% (Table 5). Despite its very low amount in AO (Table 1), its levels were multiplied by 15 with respect to control milk. These results could be a consequence of the retroconversion of DHA to EPA in the tissues, previously observed by Alvarenga et al. [54] when lambs were fed with algae.

Based on the milk fat content reported in Table 2, the calculated sum of EPA *plus* DHA in the AO experimental group rendered a total of 147 mg/100 g of milk. According to the European Union regulations [55], which define a food ‘*high in omega-3 fatty acids*’ if it contains 80 mg of EPA + DHA *per* 100 g food, the inclusion of 2.3% AO in the assayed experimental diets would allow this standard to be met in ewe milk. In comparison, milk from animals fed LO and CaS-LO diets provided barely 48 and 38 mg of ALA, respectively, per 100 g of milk. These values are very far from the claim that a food is ‘*high in omega-3 fatty acids*’ or even a ‘*source of omega-3 fatty acids*’. These claims may only be made when the product contains at least 0.6 (*high*) or 0.3 g (*source*) of ALA *per* 100 g of food [55]. It is therefore evident that AO supplementation in the present trial was the most efficient way to contribute towards meeting n-3 FA requirements.

The percentages of DPA in AO diet milk were six times higher than in the control diet, whereas control and LO supplemented diets did not statistically differ (Table 5). Although DPA is not considered in the regulations as a source of health-claimable n-3 FA [55], different beneficial properties have been reported in human nutrition. Among other effects, DPA is the precursor for oxylipins and anti-inflammatory and neuroprotective compounds, and it stimulates endothelial cell migration much more efficiently than EPA [1,52]. Thus, the presence of DPA in ewe’s milk would enhance the actual n-3 FA content to higher values and could improve its composition from a nutritional perspective.

The ratio n-6/n-3 is generally considered to be essential when judging the nutritional composition of foods from the point of view of human health. It is desirable that the ratio is below a maximum recommended level of four [56]. In the present research, the lowest n-6/n-3 ratio (0.65) was obtained in milk derived from the AO diet and reflects the greater dietary concentration and tissue uptake of long-chain n-3 PUFA in this treatment. The use of LO supplemented diets (LO and CaS-LO) also decreased this ratio but algae supplementation was demonstrated to be more effective (Table 5). The lowest thrombogenic index (TI) was also obtained for AO samples (1.82), reduced by half in comparison to control, which evidences the nutritional advantage of AO supplemented diet milk. In contrast, the lowest values of the atherogenic index were observed in samples derived from linseed oil supplemented diets (Table 5).

## 4. Conclusions

In dairy ewes, the replacement of a fat enriched in SFA by 2.3% with fat from different sources of n-3 FA (linseed oil, calcium salts of linseed oil and algae oil) contributed to increasing n-3 FA, VA and RA levels in milk and to decreasing the content of SFA in different degrees. In relation to health implications, AO was an effective strategy to enrich ewe milk with DPA, EPA and DHA and to decrease the TI index and the n6/n3 ratio. This effect may have important implications for the manufacture of dairy products with a healthier FA profile. However, ruminal protected forms of AO should be considered in order to prevent its negative effects on DMI, milk yield and quality. The few differences observed between LO and CaS-LO in the milk FA profile reject the hypothesis that supplementation with CaS-LO could avoid the BH of n-3 FA, such as ALA, in the rumen. Finally, the consequences that may arise, in terms of oxidative stability, when ewes’ milk cheese has a high content of long-chain n-3 PUFA deserve to be studied further

## Figures and Tables

**Table 1 animals-12-01716-t001:** Ingredients and chemical composition of the experimental diets.

	Diet ^1^			
Item	Control	LO	CaS-LO	AO
Ingredients (g/100 g of fresh matter)				
Dehydrated alfalfa hay	35.0	35.0	34.8	35.0
Barley straw	9.0	9.0	8.9	9.0
Soybean meal	14.0	14.0	13.9	14.0
Whole corn grain	10.4	10.4	10.3	10.4
Oat grain	9.0	9.0	8.9	9.0
Whole barley grain	8.0	8.0	8.0	8.0
Beet pulp	7.0	7.0	7.0	7.0
Molasses	4.3	4.3	4.3	4.3
Hydrogenated palm oil ^2^	2.3	-	-	
Linseed oil ^3^	-	2.3	-	-
Calcium salts of linseed oil ^4^	-	-	2.9	-
Algae oil ^5^	-	-	-	2.3
Vitamin mineral premix	1.0	1.0	1.0	1.0
Chemical composition (g/100 g of DM)				
Organic matter	93.1	92.9	93.3	93.9
Crude protein	17.4	17.3	17.2	17.3
Ether extract	4.6	4.5	4.6	4.5
NDF	34.1	34.3	34.8	34.5
ADF	22.3	22.5	22.1	22.6

^1^ Diets: Control = TMR supplemented with 2.3% of hydrogenated palm oil (PROFAT, Mateos S.L.); LO = TMR supplemented with 2.3% of linseed oil (ECOFLAX FEED OMEGA-3, BTSA); CaS-LO = TMR supplemented with 2.3% of fat from calcium salts of linseed oil (LINOFAT, Nutrion International SLU); AO = TMR supplemented with 2.3% of algae oil (BIOMEGA TECH A 40 FEED, BTSA). ^2^ Contained (g/100 g total fatty acids): 12:0 (0.12), 14:0 (1.3), 16:0 (66.2), 16:1 (<0.1), 18:0 (31.0), *cis*-9 18:1 (<0.1). ^3^ Contained (g/100 g total fatty acids): 16:0 (5.2), 18:0 (3.5), *cis*-9 18:1 (21.1), 18:2 (14.4), 18:3 (55.5). ^4^ Contained 805 g of total fatty acids/kg and (g/100 g total fatty acids): 16:0 (5.2), 18:0 (3.2), *cis*-9 18:1 (18.4), 18:2 (15.0), 18:3 (58.0). ^5^ Contained (g/100 g total fatty acids): 14:0 (9.0), 16:0 (19.3), *cis*-9 16:1 (5.3), 18:0 (0.7), *cis*-9 18:1 (7.8), 18:2 (0.4), 18:3 (0.2), EPA (1.7), DPA (7.8), DHA (43.8).

**Table 2 animals-12-01716-t002:** Dry matter intake (DMI), and milk yield and composition in ewes fed a TMR containing 2.3% of different lipid supplements.

	Diet ^1^					*p*-Value ^3^		
Item	Control	LO	CaS-LO	AO	SED ^2^	D	T	D × T
DMI (g/d)	2691 ^a^	2759 ^a^	2736 ^a^	1601 ^b^	80.9	<0.001	<0.001	0.847
Yield (g/d)								
Milk	1359 ^a^	1376 ^a^	1435 ^a^	1013 ^b^	58.0	<0.001	<0.001	0.398
Fat	72.5 ^b^	79.4 ^ab^	80.4 ^a^	51.2 ^c^	3.59	<0.001	<0.001	0.196
Protein	69.8 ^a^	70.1 ^a^	73.0 ^a^	52.4 ^b^	2.76	<0.001	<0.001	0.335
Lactose	68.1 ^a^	68.8 ^a^	72.5 ^a^	48.3 ^b^	3.05	<0.001	<0.001	0.436
Total solids	222.6 ^a^	230.7 ^a^	238.8 ^a^	161.0 ^b^	9.54	<0.001	<0.001	0.319
Composition (g/100 g)								
Fat	5.32 ^b^	5.69 ^a^	5.61 ^a^	5.11 ^b^	0.12	<0.001	<0.001	0.540
Protein	5.19 ^b^	5.11 ^b^	5.12 ^b^	5.35 ^a^	0.08	0.010	0.003	0.012
Lactose	4.99 ^a^	4.97 ^a^	5.03 ^a^	4.69 ^b^	0.04	<0.001	0.002	0.001
Total solids	16.29 ^a^	16.68 ^a^	16.67 ^a^	16.04 ^b^	0.15	<0.001	0.006	0.270

^1^ Diets: Control = TMR supplemented with 2.3% of hydrogenated palm oil (PROFAT, Mateos S.L, Valladolid, Spain); LO = TMR supplemented with 2.3% of linseed oil (ECOFLAX FEED OMEGA-3, BTSA, Alcalá de Henares, Madrid, Spain); CaS-LO = TMR supplemented with 2.3% of fat from calcium salts of linseed oil (LINOFAT, Nutrion International SLU, Madrid, Spain); AO = TMR supplemented with 2.3% of algae oil (BIOMEGA TECH A 40 FEED, BTSA, Alcalá de Henares, Madrid, Spain). ^2^ Standard error of the difference. ^3^ Probability of significant effect of lipid supplement (D), week of lactation during the experimental period (T), and their interaction (D × T). ^a–c^ Means with different superscripts differ significantly (*p* < 0.05).

**Table 3 animals-12-01716-t003:** Saturated fatty acid (SFA) profile (g/100 g of total fatty acid methyl esters) of milk fat from ewes fed with 2.3% of different lipid supplements.

	Diets ^1^					Probability ^3^		
Item	Control	LO	CaS-LO	AO	SED ^2^	D	T	D × T
4:0	3.47 ^b^	3.93 ^a^	3.81 ^a^	3.16 ^c^	0.137	<0.001	0.261	0.138
5:0	0.02 ^a^	0.03 ^a^	0.03 ^a^	0.02 ^b^	0.003	0.001	0.267	0.680
6:0	3.19 ^b^	3.31 ^ab^	3.39 ^ab^	3.49 ^a^	0.126	0.119	0.117	0.429
7:0	0.07 ^a^	0.05 ^b^	0.05 ^b^	0.05 ^b^	0.007	0.014	0.248	0.324
8:0	3.23 ^b^	3.19 ^b^	3.31 ^b^	3.77 ^a^	0.153	0.001	0.092	0.800
9:0	0.12 ^a^	0.08 ^b^	0.09 ^b^	0.08 ^b^	0.011	0.001	0.376	0.265
10:0	11.82 ^a^	9.15 ^c^	10.30 ^b^	10.90 ^ab^	0.499	<0.001	0.348	0.285
11:0	0.16 ^a^	0.07 ^b^	0.10 ^b^	0.07 ^b^	0.022	<0.001	0.189	0.154
12:0	7.74 ^a^	4.86 ^c^	5.69 ^b^	5.83 ^b^	0.382	<0.001	0.864	0.018
13:0	0.13 ^a^	0.07 ^b^	0.08 ^b^	0.08 ^b^	0.012	<0.001	0.084	0.291
14:0	13.97 ^a^	10.81 ^c^	11.59 ^b^	12.19 ^b^	0.381	<0.001	0.010	0.572
15:0	1.13 ^a^	0.77 ^b^	0.82 ^b^	0.80 ^b^	0.054	<0.001	0.046	0.488
16:0	26.88 ^a^	19.81 ^c^	21.25 ^b^	27.27 ^a^	0.705	<0.001	0.346	0.071
17:0	0.71 ^a^	0.47 ^c^	0.52 ^bc^	0.54 ^b^	0.023	<0.001	0.001	0.034
18:0	4.57 ^c^	7.97 ^a^	6.69 ^b^	1.31 ^d^	0.484	<0.001	0.698	0.991
19:0	0.05 ^b^	0.09 ^a^	0.09 ^a^	0.06 ^b^	0.006	<0.001	0.467	0.109
20:0	0.14 ^c^	0.19 ^a^	0.16 ^b^	0.10 ^d^	0.009	<0.001	0.117	0.475
21:0	0.04 ^b^	0.14 ^a^	0.10 ^a^	0.04 ^b^	0.014	<0.001	0.474	0.639
22:0	0.05 ^b^	0.06 ^a^	0.06 ^a^	0.04 ^b^	0.004	<0.001	0.708	0.110
23:0	0.02 ^c^	0.02 ^bc^	0.02 ^b^	0.03 ^a^	0.002	<0.001	0.216	0.003
24:0	0.01 ^b^	0.01 ^b^	0.01 ^b^	0.04 ^a^	0.002	<0.001	0.996	0.186
TOTAL SFA (non-branched)	77.52 ^a^	65.07 ^c^	68.15 ^b^	69.87 ^b^	0.974	<0.001	0.447	0.806
*iso* 13:0	0.01 ^c^	0.01 ^b^	0.01 ^c^	0.02 ^a^	0.001	<0.001	0.022	0.994
*anteiso* 13:0	0.11 ^a^	0.06 ^b^	0.05 ^b^	0.05 ^b^	0.008	<0.001	0.007	0.186
*iso* 14:0	0.08 ^a^	0.07 ^ab^	0.06 ^c^	0.07 ^bc^	0.005	0.007	0.004	0.000
*iso* 15:0	0.13 ^c^	0.15 ^b^	0.13 ^c^	0.19 ^a^	0.009	<0.001	0.009	0.051
*anteiso* 15:0	0.35 ^b^	0.33 ^bc^	0.31 ^c^	0.38 ^a^	0.017	0.001	0.632	0.044
*iso* 16:0	0.23	0.22	0.21	0.24	0.021	0.714	0.087	0.404
*iso* 17:0	0.16 ^c^	0.21 ^b^	0.18 ^c^	0.27 ^a^	0.015	<0.001	0.363	0.972
*anteiso* 17:0	0.38	0.28	0.27	0.27	0.062	0.254	0.644	0.149
*iso* 18:0	0.05 ^b^	0.04 ^c^	0.04 ^c^	0.06 ^a^	0.003	<0.001	0.235	0.221
TOTAL SFA (branched)	1.49 ^ab^	1.37 ^bc^	1.26 ^c^	1.55 ^a^	0.085	0.005	0.185	0.170
TOTAL SFA	79.00 ^a^	66.44 ^d^	69.41 ^c^	71.42 ^b^	0.976	<0.001	0.521	0.813
HSFA ^4^	48.58 ^a^	35.48 ^d^	38.53 ^c^	45.29 ^b^	0.898	<0.001	0.055	0.920

^1^ Diets: Control = TMR supplemented with 2.3% of hydrogenated palm oil (PROFAT, Mateos S.L., Valladolid, Spain); LO = TMR supplemented with 2.3% of linseed oil (ECOFLAX FEED OMEGA-3, BTSA, Alcala de Henares, Madrid, Spain); CaS-LO = TMR supplemented with 2.3% of fat from calcium salts of linseed oil (LINOFAT, Nutrion International SLU, Madrid, Spain); AO = TMR supplemented with 2.3% of algae oil (BIOMEGA TECH A 40 FEED, BTSA, Alcala de Henares, Madrid, Spain). ^2^ Standard error of the difference. ^3^ Probability of significant effect of lipid supplement (D), time on diet (T), and their interaction (D × T). ^a–d^ Means with different superscripts differ significantly (*p* < 0.05). ^4^ The hypercholesterolemic saturated fatty acids, HSFA = 12:0 + 14:0 + 16:0 [38].

**Table 4 animals-12-01716-t004:** Monounsaturated fatty acid (MUFA) profile (g/100 g of total fatty acid methyl esters) of milk fat from ewes fed with 2.3% of different lipid supplements.

	Diets ^1^					Probability ^3^		
Item	Control	LO	CaS-LO	AO	SED ^2^	D	T	D × T
10:1	0.51 ^a^	0.37 ^b^	0.32 ^b^	0.37 ^b^	0.031	<0.001	0.025	0.690
*cis*-11 12:1	0.23 ^a^	0.10 ^b^	0.11 ^b^	0.10 ^b^	0.019	<0.001	0.042	0.065
*cis*-9 14:1	0.37 ^a^	0.23 ^b^	0.21 ^bc^	0.17 ^c^	0.030	<0.001	0.000	0.245
*cis*-7 16:1	0.23 ^a^	0.23 ^a^	0.22 ^ab^	0.20 ^b^	0.014	0.075	0.112	0.335
*cis*-8 16:1	0.03 ^b^	0.06 ^a^	0.05 ^a^	0.05 ^a^	0.007	0.002	0.155	0.569
*cis*-9 16:1	1.09 ^a^	0.77 ^b^	0.71 ^b^	0.95 ^a^	0.070	<0.001	0.051	0.949
Other *cis* 16:1	0.08	0.11	0.12	0.17	0.646	0.534	0.3947	0.488
*cis*-13 16:1	0.02 ^b^	0.03 ^a^	0.02 ^a^	0.02 ^b^	0.002	<0.001	0.760	0.296
*cis*-9 17:1	0.17 ^a^	0.12 ^b^	0.13 ^b^	0.09 ^c^	0.009	<0.001	0.917	0.277
*cis*-9 18:1	11.46 ^c^	15.37 ^a^	13.17 ^b^	5.97 ^d^	0.623	<0.001	0.093	0.794
*cis*-11 18:1	0.48 ^b^	0.54 ^b^	0.55 ^b^	1.07 ^a^	0.038	<0.001	0.207	0.353
*cis*-12 18:1	0.19 ^b^	0.53 ^a^	0.58 ^a^	0.07 ^c^	0.047	<0.001	0.003	0.136
*cis*-13 18:1	0.02 ^b^	0.04 ^a^	0.04 ^a^	0.02 ^b^	0.003	<0.001	0.433	0.125
*cis*-15 18:1	0.04 ^b^	0.25 ^a^	0.19 ^a^	0.03 ^b^	0.032	<0.001	0.446	0.684
*cis*-11 20:1	0.07 ^c^	0.09 ^b^	0.10 ^a^	0.09 ^ab^	0.005	<0.001	0.125	0.308
Other 20:1	0.04 ^c^	0.07 ^a^	0.06 ^b^	0.06 ^b^	0.004	<0.001	0.006	0.129
22:1 n-9	0.01 ^c^	0.01 ^b^	0.01 ^bc^	0.01 ^a^	0.001	<0.001	0.069	0.676
TOTAL MUFA *cis*	15.05 ^c^	18.93 ^a^	16.61 ^b^	9.44 ^d^	0.651	<0.001	0.218	0.803
*trans* 15:1	0.09 ^b^	0.09 ^b^	0.09 ^b^	0.14 ^a^	0.008	<0.001	0.051	0.858
*trans*-5 16:1	0.02 ^a^	0.02 ^b^	0.02 ^b^	0.01 ^b^	0.002	<0.001	0.154	0.083
*trans*-6/7 16:1	0.02	0.02	0.02	0.02	0.001	0.699	0.039	0.816
*trans*-8 16:1	0.06 ^b^	0.10 ^a^	0.09 ^a^	0.09 ^a^	0.006	<0.001	0.346	0.544
*trans*-9 16:1	0.11 ^c^	0.31 ^b^	0.26 ^b^	0.48 ^a^	0.032	<0.001	0.758	0.924
*trans*-10 16:1	0.03 ^b^	0.05 ^b^	0.04 ^b^	0.08 ^a^	0.010	<0.001	0.818	0.731
*trans*-11 + *trans*-12 16:1	0.02 ^c^	0.09 ^a^	0.09 ^a^	0.06 ^b^	0.005	<0.001	0.932	0.981
*trans*-4 18:1	0.01 ^d^	0.02 ^b^	0.03 ^a^	0.00 ^c^	0.002	<0.001	0.082	0.273
*trans*-5 18:1	0.02 ^d^	0.02 ^b^	0.02 ^a^	0.02 ^bc^	0.002	<0.001	0.295	0.041
*trans*-6+*trans*-7+*trans*-8 18:1	0.13 ^b^	0.39 ^a^	0.39 ^a^	0.11 ^b^	0.022	<0.001	0.169	0.080
*trans*-9 18:1	0.15 ^c^	0.40 ^a^	0.39 ^a^	0.24 ^b^	0.025	<0.001	0.310	0.556
*trans*-10 18:1	0.25 ^b^	0.77 ^a^	0.69 ^ab^	0.66 ^ab^	0.235	0.128	0.067	0.335
*trans*-11 18:1	0.42 ^c^	3.55 ^b^	3.55 ^b^	5.20 ^a^	0.423	<0.001	0.169	0.481
*trans*-12 18:1	0.23 ^c^	0.78 ^a^	0.80 ^a^	0.65 ^b^	0.045	<0.001	0.172	0.138
*trans*-16 18:1 + *cis*-14 18:1	0.16 ^b^	0.43 ^a^	0.39 ^a^	0.06 ^c^	0.025	<0.001	0.479	0.944
TOTAL MUFA *trans*	1.71 ^b^	7.04 ^a^	6.89 ^a^	7.85 ^a^	0.517	<0.001	0.582	0.341
TOTAL MUFA	16.76 ^c^	25.97 ^a^	23.50 ^b^	17.29 ^c^	0.773	<0.001	0.161	0.878

^1^ Diets: Control = TMR supplemented with 2.3% of hydrogenated palm oil (PROFAT, Mateos S.L. Valladolid, Spain); LO = TMR supplemented with 2.3% of linseed oil (ECOFLAX FEED OMEGA-3, BTSA, Alcala de Henares, Madrid, Spain); CaS-LO = TMR supplemented with 2.3% of fat from calcium salts of linseed oil (LINOFAT, Nutrion International SLU, Madrid, Spain); AO = TMR supplemented with 2.3% of algae oil (BIOMEGA TECH A 40 FEED, BTSA, Alcala de Henares, Madrid, Spain). ^2^ Standard error of the difference. ^3^ Probability of significant effect of lipid supplement (D), time on diet (T), and their interaction (D × T). ^a–d^ Means with different superscripts differ significantly (*p* < 0.05).

## Data Availability

The data presented in this study are available in the article.

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
