# Peer review of "Enrichment of Ewe’s Milk with Dietary n-3 Fatty Acids from Palm, Linseed and Algae Oils in Isoenergetic Rations"

_animals, 2022, doi:10.3390/ani12131716_

Round 1

Reviewer 1 Report

The manuscript is well written, and the information provided in the paper is suitable for the aims and scope of the journal. In my view, the work reported in this paper has already available in the literature from previous studies conducted in dairy animals or meat producing ruminants. In this space the paper does not report anything innovative or new findings. However, I have some major points to clarify before the acceptance of paper. I would like the authors addressing those points before publication of this manuscript.

Reviewer 2 Report

Dear authors,

Thank you for the interesting article. My concerns are minor and they are indicated in the text attached.

The paper examines the possibilities of enriching milk with the dietary addition of different fatty acids. The topic is very relevant since there is a growing demand for healthy milk production worldwide. The paper presents novel and valuable findings. The introduction provides evidence-based background for the research. The methods have been adequately described, results are well presented and data interpretation is appropriate. The findings are thoroughly discussed, and conclusions are justified by the results. I did not find any factual errors. 

All the best and stay safe,

Author Response

1. Change the title in:  Enrichment of ewe`s milk with dietary n-3 fatty acids from palm, linseed and algae oils in isoenergetic rations

We have modified the title according to your suggestion.

2. Insert: DOI: https://doi.org/10.1017/S0043933914000634

Thanks for the reference, we have included it in the new version of the manuscript.

Reviewer 3 Report

The manuscript describes a well-implemented experimental activity. However, there are some critical points that need to be addressed.

Line96: you put 23 g/100 g, but I suppose that authors mean 2.3 g/100 g

Actually, in notes of Table 1 the authors indicate that in the CaS-LO diet they added 2.3 percent of LINOFAT, but in the table they report 2.9.

in notes of table1, please, check the values reported for fatty acid composition of control diet, the sum of values is too different from 100, and correct C18.3 with C18:3.

I suggest carefully checking the values given in the tables to make them consistent with the digits.

In the m&m session, the authors reported that they made the measurements on the animals on a weekly/biweekly basis during the experimental period. However, there is no information in the presentation of the results that allow to understand, at least when the effect of time was significant, how the observed items evolved along the experiment. Line96: you put 23 g/100 g, but I suppose that authors mean 2.3 g/100 g

Actually, in notes of Table 1 the authors indicate that in the CaS-LO diet they added 2.3 percent of LINOFAT, but in the table they report 2.9.

in notes of table1, please, check the values reported for fatty acid composition of control diet, the sum of values is too different from 100, and correct C18.3 with C18:3.

I suggest carefully checking the values given in the tables to make them consistent with the digits.

In the m&m session, the authors reported that they made the measurements on the animals on a weekly/biweekly basis during the experimental period. However, there is no information in the presentation of the results that allow to understand, at least when the effect of time was significant, how the observed items evolved along the experiment.

# lines 223-226: changes in milk composition, such as decreased lactose and increased protein concentration, are observed in ewes with subclinical mastitis. The authors have information on the udder health status of lactating ewes, like SSC in milk?

# lines 256-257: I suggest to rephrase the sentence, because the effect is due to the replacement of SFA with n-3 FA in the diets. This aspect must be considered throughout the presentation of the results and in their discussion, since in this experiment the "control" is a diet enriched with saturated fatty acids.

# lines 390-392: can the authors provide an explanation for the lack of effect of linseed oil in the diet, compared with the SFA-rich diet, on the content of long-chain n-3 FA in milk?

# lines 418-419: Explain what calculations were done to quantify the transfer efficiency of long-chain FAs from diet in to milk.

# lines 453-454: Please rephrase the sentence, actually with the experiment it was not supplemented the diet, but a source of SFA was replaced with sources with a different kind of n-3 PUFA.

# conclusions: I suggest to add some thoughts about the consequences that may arise, in terms of oxidative stability of the product, when the cheese has high content of long-chain n-3 PUFAs.

Round 2

Reviewer 3 Report

Authors have improved the manuscript and well justified their choices. Therefore, I propose to accept the manuscript for publication